# Prediction of Hemorrhagic Complication after Thrombolytic Therapy Based on Multimodal Data from Multiple Centers: An Approach to Machine Learning and System Implementation

**DOI:** 10.3390/jpm12122052

**Published:** 2022-12-12

**Authors:** Shaoguo Cui, Haojie Song, Huanhuan Ren, Xi Wang, Zheng Xie, Hao Wen, Yongmei Li

**Affiliations:** 1School of Computer and Information Science, Chongqing Normal University, Chongqing 401331, China; 2Department of Radiology, the First Affiliated Hospital of Chongqing Medical University, Chongqing 400016, China; 3Department of Radiology, Chongqing General Hospital, Chongqing 400013, China; 4School of Economics, Peking University, Beijing 100871, China

**Keywords:** hemorrhagic complication, machine learning, XGB, clinical decision support system

## Abstract

Hemorrhagic complication (HC) is the most severe complication of intravenous thrombolysis (IVT) in patients with acute ischemic stroke (AIS). This study aimed to build a machine learning (ML) prediction model and an application system for a personalized analysis of the risk of HC in patients undergoing IVT therapy. We included patients from Chongqing, Hainan and other centers, including Computed Tomography (CT) images, demographics, and other data, before the occurrence of HC. After feature engineering, a better feature subset was obtained, which was used to build a machine learning (ML) prediction model (Logistic Regression (LR), Random Forest (RF), Support Vector Machine (SVM), eXtreme Gradient Boosting (XGB)), and then evaluated with relevant indicators. Finally, a prediction model with better performance was obtained. Based on this, an application system was built using the Flask framework. A total of 517 patients were included, of which 332 were in the training cohort, 83 were in the internal validation cohort, and 102 were in the external validation cohort. After evaluation, the performance of the XGB model is better, with an AUC of 0.9454 and ACC of 0.8554 on the internal validation cohort, and 0.9142 and ACC of 0.8431 on the external validation cohort. A total of 18 features were used to construct the model, including hemoglobin and fasting blood sugar. Furthermore, the validity of the model is demonstrated through decision curves. Subsequently, a system prototype is developed to verify the test prediction effect. The clinical decision support system (CDSS) embedded with the XGB model based on clinical data and image features can better carry out personalized analysis of the risk of HC in intravenous injection patients.

## 1. Introduction

Stroke is the second leading cause of death globally and a leading cause of disability [1,2,3]. Acute ischemic stroke (AIS) is the most common type of stroke, accounting for about 80% to 87%; it ranks first in the stroke rankings and has the characteristics of high morbidity, high disability rate, high fatality rate, and high recurrence rate [4]. Intravenous thrombolytic (IVT) therapy with drugs is beneficial in patients with AIS within 4.5 h of stroke onset [5,6]. However, hemorrhagic complication (HC) is the most severe complication of IVT and is associated with poor clinical outcomes [7,8,9].

Currently, machine learning (ML), a branch of artificial intelligence, provides a promising tool in the pursuit of personalized outcome prediction, which is widely used in medical practice as a clinical decision support system (CDSS) [3,9,10,11,12,13,14,15]. To improve personalized stroke care, Hu et al. developed and validated a dynamic ML model based on eXtreme Gradient Boosting (XGB) at multiple time points for dynamically predicting adverse outcomes at three months [3]. Van Os HJA et al. used machine learning methods such as Random Forest (RF) and Support Vector Machine (SVM) to predict the outcome of endovascular treatment for AIS [13].

In several of these studies, the ML algorithm was applied to predict HC risk after AIS thrombolytic therapy [16,17,18,19]. Choi JM et al. constructed multiple ML models to predict HC, and the AUC of the artificial neural network in predicting HC after thrombolysis in AIS patients reached 0.844 [16]. In another experiment, multivariate Logistic Regression (LR) analysis showed an AUC of 0.776 for predicting HC on the test set [17]. The above two studies only used demographic, laboratory tests and other clinical data and did not include any image information. Miao LQ et al. used radiomics to build a prediction model based on an SVM classifier that predicted HC on the test set with an AUC of 0.921 [18]; In this study, only image features were used, and clinical information was not considered. Liu Z et al. reached an AUC of 0.885 for the prediction of HC based on multimodal magnetic resonance imaging omics and clinical risk factors by using LR [19]. Although Liu Z used multimodal data, it has the problem of too few samples and no external verification. It can be seen from the above that it is feasible to use imaging features and clinical factors to establish an ML model to predict HC risk. However, the current research still has some limitations. In addition, the ML model is similar to a black box, and users do not know how the model gets results. The above research did not consider the interpretability of the model and did not develop an application system, which is not conducive to the practical application and promotion of the prediction model.

In this study, we developed machine learning models based on data from multiple centers and explored multiple modality data of image and clinical information. We solved single data modality, few samples, and no external verification problems, improving the model’s generalization ability further. In addition, we used the tool kit to analyze the interpretability of the model and developed a CDSS based on the prediction model. As far as we know, this is the first study to propose and build a prediction system for HC after thrombolytic treatment of AIS. The final prediction model and system are helpful for a personalized analysis of the risk of HC in patients receiving thrombolytic therapy to provide reference information for doctors and bring an excellent prognosis to patients.

## 2. Materials and Methods

### 2.1. Data Information

This retrospective study was approved by the Institutional Review Board of the Affiliated Hospital of Chongqing Medical University, exempting patients from the requirement for informed consent. We conducted a multicenter study, including data from the Affiliated Hospital of Chongqing Medical University, the Second People’s Hospital of Hunan Province, the Second Hospital of Ledong County, Changsha Central Hospital, the First Affiliated Hospital of Hainan Medical College, and the People’s Hospital of Haikou City. The inclusion criterion was patients who received IVT under the guidelines for the management of AIS and underwent non-contrast computed tomography (NCCT) before IVT treatment. Patients with AIS were included in this study if follow-up Magnetic Resonance Imaging (MRI) or NCCT was performed within 36 h after IVT treatment. The following patients were excluded from this study: patients with severely missing data; patients with severe artifacts on NCCT images; patients with primary intracerebral hemorrhage, brain tumor, head trauma, and hemorrhagic infarction on admission.

All patient data and radiographic images were manually extracted from electronic medical records using standardized case report forms, including: demographics (age, gender, smoking history, drinking history (The description of smoking and drinking habits uses the Smoking index [20] and Drinking index [21])), medical history (stroke history, diabetes history, atrial fibrillation history), clinical manifestations (time of onset, systolic blood pressure, diastolic blood pressure, National Institutes of Health Stroke Scale (NIHSS) score at admission, TOAST classification), laboratory examinations (hemoglobin, platelets, leukocytes, neutrophils, lymphocytes, monocytes, neutrophil-to-lymphocyte ratio, lymphocyte-to-monocyte ratio, absolute value of eosinophils, total cholesterol, triglycerides, high-density lipoprotein, low-density lipoprotein, Blood Urea Nitrogen (BUN), creatinine, Rapid blood glucose, glycated hemoglobin, international normalized ratio (INR), plasma fibrin degradation products (FDP), D-dimer (dynamic), Brain Natriuretic Peptide (BNP) precursor) and patient NCCT images. ITK-SNAP (https://itk.org, accessed on 24 August 2022) extracts the radiologist imaging features, which are used after delineating the region of interest. ITK-SNAP is an open-source software that mainly includes histogram information, morphological features, co-occurrence matrix feature parameters, run-length matrix features, gray-scale connected region matrix, and other features information.

### 2.2. Machine Learning Algorithms

The overall flow of model establishment is shown in Figure 1. The data is divided into a training set, an internal validation data set, and external validation data set. Then, patients’ CT images are delineated, features are extracted, and demographics and other information are combined. Next, feature engineering is carried out to achieve the purpose of dimensionality reduction, and then four prediction models are constructed. We use criterions such as ACC and AUC to evaluate the performance of models. Finally, the prediction model with better performance and the variables included in the model are obtained.

#### 2.2.1. Data Processing

The data from Haikou People’s Hospital was treated as the external validation group, and the data from other hospitals were randomly divided into a training group and an internal validation group according to a ratio of 8:2. In addition, missing values are filled by the MissForest algorithm. Each data queue is filled separately to avoid data leakage. MissForest algorithm can fill continuous and categorical variables simultaneously, and studies have found that its filling effect is significantly better than methods such as K-Nearest Neighbor (KNN) imputation and multiple imputation (MI) [22]. MI method considers the data distribution of missing variables, estimates the distribution of missing values, and extracts several (≥2) data to fill the missing values. For feature selection, first of all, based on the repeated delineation of the patient images of the First Affiliated Hospital of Chongqing Medical University by two radiologists, the two groups of feature data were obtained from consistency analysis. The filtered features were applied to the training data according to the threshold value to reduce the effect of edge features on prediction and ensure that the features have high repeatability [23]. We standardize the data to have zero mean and unit variance. Then, clinical data (demographics, past medical history, clinical manifestations, and laboratory indicators collectively) were combined with imaging features. After univariate analysis, features were further screened by Lasso [24] for downstream machine-learning models.

#### 2.2.2. Algorithm Model

Machine learning algorithms used in this research are LR, RF, SVM, and XGB, which are widely used and successful at present [3,10,11,12,13]. We obtained the best-performing predictive model by comparing the model’s evaluation with the validation dataset. In the experiment, we used the scikit-learn module in Python to implement the machine-learning algorithms [25]. This article uses Python 3.7 programming, modeling, and training on the Windows 10 (21H2) operating system configuration, the CPU is AMD Ryzen 7 5800 H with Radeon Graphics, 3.20 GHz, and the memory is 32 G.


Logistic Regression


LR is a commonly used a linear model that can be used to solve classification problems. Recently, it has been widely used as a medical prediction model because it is simple and effective and can draw a nomogram, which is intuitive and convenient for practical application [19]. You can learn more about the LR algorithm in [26] (p. 73–88).


Random Forest


RF plays an important role in disease prediction and diagnosis and is widely used; it is an ensemble classifier composed of multiple decision tree classifiers. RF includes several (random) decision trees. When the number becomes larger, the generalization error of the forest converges to a limit [27,28]. Each decision tree classifier decides the optimal classification result by voting. The classification idea of random forest is in [29].


Support Vector Machine


SVM belongs to a binary classification model. The basic idea of the algorithm is to map the input feature space of the sample into a high-dimensional feature space and find the optimal classification hyper-plane that meets the classification requirements in this high-dimensional feature space, so that the plane has a certain classification accuracy. The algorithm can better solve practical problems such as nonlinearity and small samples and has high classification and recognition accuracy and strong generalization ability. It has been widely used in big data mining, image recognition, medical detection, and other fields [30,31,32]. Details can be found in [33].


eXtreme Gradient Boosting


XGB is an improvement of the gradient-boosting decision tree (GBDT) method, which can be used for classification and regression problems. It is also a boosting iterative algorithm that integrates many weak classifiers to form a strong classifier [34,35]. Each iteration of the algorithm will generate a tree to learn the deviation between the predicted value of the model and the actual value. In recent years, it has been widely used in the field of machine learning because of its superior performance [36]. The workflow of the algorithm can be seen in [35].

### 2.3. Statistical Analysis

Statistical analysis was performed using Python software (version 3.7) and R (version 4.1.3). Differences in the baseline characteristics of the datasets were assessed with one-way ANOVA or the nonparametric Kruskal-Wallis test. The Delong test was used to compare the prediction ability between different prediction models. In addition, the Hosmer-Lemeshow test was performed to compare the degree of agreement between predicted and observed probabilities. We set a two-sided *p* < 0.05 to be considered statistically significant.

We calculated Accuracy (ACC), Sensitivity (SEN), Specificity (SPE), Positive Predictive Value (PPV), Negative Predictive Value (NPV), Receiver Operating Characteristic (ROC) curves and the AUC values and F1 values as an evaluation metric, the performance of the ML algorithm is compared between the training set and the test set, and the best performing model is selected. For model interpretation, based on The SHapley Additive exPlanation (SHAP) framework calculates the Shapley value for each variable to further explain the best-performing model, providing support for the reliability of the model results [36,37].

### 2.4. Prototype System Construction

We adopted the Browser/Server (B/S) architecture and used a python flask and its related extensions to design and implement the clinical decision support system. Get the parameters required by the model through the webpage and display the predicted risk results of the model on the resulting interface. In addition, we take the waterfall diagram in the SHAP framework as the interpretability analysis of the prediction results, which can clarify the impact of characteristics on patients.

## 3. Results

### 3.1. Statistical Results

Finally, 517 patients were enrolled in the study, with an average age of 67.02 ± 12.67 years, including 333 males and 184 females, accounting for 64.4% and 35.6%, respectively. The patients were divided into the training cohort (332), internal validation cohort (83) and external validation cohort (102). The comparison of baseline characteristics between training data and internal test data is shown in Table 1; it includes the variables finally included, basic information, and variables with significant differences between the two data sets. The comparison of baseline characteristics between external validation data and training data and internal validation data is shown in Appendix A. There is no significant difference in most indicators, but there are differences in several indicators such as atrial fibrillation history, systolic blood pressure, and diastolic blood pressure.

### 3.2. Feature Screening Results

First, ICC analysis is carried out on the 1073-dimensional image feature based on the repeated delineation data, and 843-dimensional image features are obtained according to the threshold value of 0.94. The selection of the threshold value is shown in Appendix A. Through univariate analysis, the obtained image features and 33-dimensional clinical features were analyzed together, and the indicators that conform to the normal distribution were tested by a bilateral T-test. Other indicators were analyzed by the non-parametric Mann-Whitney U-test to see if there was a significant difference, and 838-dimensional features were obtained. The Lasso method is further used to reduce the dimensionality, and the features whose corresponding coefficient to the Lambda coefficient is not 0 are selected to obtain 66-dimensional features. There are 15 clinical features and 51 image features. The change curve of loss and characteristic coefficient Lambda is shown in Figure 2. The features of the weight Top 30 are shown in Figure 3.

Before building the four prediction models, the 66-dimensional features are sorted according to the absolute value of the feature weight, input into each model in turn, and select the appropriate number of features through the performance of different quantity features in the verification set of 10-fold cross-validation on the training set. The performance of different quantitative features on the four models is shown in Appendix A. Finally, the LR model is built from the first 33 features, the RF model is built from the first 24 features, the SVM model is built from the first 26 features, and the XGB model is built from the first 18 features.

### 3.3. Model Performance

10-fold cross-validation was conducted for each model on the training set (see Appendix A for the results). The performance results of the four models on each dataset are shown in Table 2 and Figure 4. The ACC and AUC of the XGB model are 0.9458 and 0.9931, respectively, on the training data, followed by the ACC and AUC of the RF model are 0.9277 and 0.9830; the ACC of the LR model on the internal validation data (0.8675) was higher than the ACC (0.8554) of the RF and XGB models, but the AUC value of the LR model was lower than the latter; on the external validation data, the XGB model delivered ACC (0.8431), AUC (0.9142), and the F1 value of 0.8222 higher than the rest of the models. Since the models performed differently on each dataset, to further compare the performance between the models, we performed a DeLong test (Table 3) for statistical analysis. From the DeLong test, it was found that the AUC performance of the XGB model on the training set was better than the other models, and the AUC performance of the four models on the internal validation and external validation cohorts had no significant difference. Overall, the XGB model has better prediction performance.

For the XGB model, the fit between the fitted values and the observed values was analyzed on the internal validation (*p* = 0.3204, Hosmer- Lemeshow test) and external validation cohort (*p* = 0.1645, Hosmer- Lemeshow test). We drew the decision curves of each model on the external validation queue (Figure 5). It can be seen that the four models have specific practical significance. The XGB model curve is almost the same as other models when the threshold probability is small, but as the threshold probability increases, the curve of the XGB model gradually surpasses other models. In addition, the prediction results are divided into three categories: high, medium, and low risk according to the third quantile of the predicted probability of the training set, there were significant differences in the three categories of data (T-test, *p* < 0.01). The proportions of high-, medium-, and low-risk in the external validation cohort were 37.25%, 43.14%, and 19.61%, respectively. From Appendix A, it can be seen that the patients classified as high-risk on the training set have all experienced bleeding transformation, while the low-risk patients have not. Some patients in the two validation sets did not meet expectations, and the proportion of medium-risk patients was higher than that in the training set. This result shows that there are many ambiguous prediction results on the validation set, and the generalization ability of the model needs to be further improved.

### 3.4. Interpretability of the Model

The idea behind SHAP feature importance is that features with large Shapley absolute values are critical. Since global importance is to be considered, the absolute Shapley values of each feature in the data are averaged, and then the features are evaluated by importance. Figure 6 shows the importance of SHAP features to the previously trained XGB model for predicting HC [38].

The summary graph (Figure 7) combines feature importance with the feature effect, and each point on the graph is the Shapley value of the feature and instance. The relationship between the eigenvalues and the predicted impact can be seen more intuitively. For example, the higher the value of rapid blood glucose, the greater the predicted probability of hemorrhagic transformation. Figure 8 is a predicted negative (low-risk) sample, the output value is lower than the base value, which is to say, the risk of this sample is lower than the overall level, and the low −0.63–(−5.07) = 5.70 is caused by different characteristics contributed.

### 3.5. Implementation of CDSS

We embedded the XGB model into the system, took predicting the risk of HC as the core task, and preliminarily designed and implemented a prototype system. Its interface is shown in Figure 9, including the system introduction, the form of input parameters, and the results display. Doctors can fill in the patient’s relevant parameter information and can get the risk analysis of HC in a few seconds. The results were quantized with probability values, and the results were interpreted with a waterfall diagram.

## 4. Discussion

In our study, we developed an easy-to-implement prediction model, which has a good prediction ability for the risk of hemorrhagic complications after thrombolysis in patients with acute ischemic stroke, and established a CDSS prototype system. The XGB model shows good prediction ability in each cohort, which helps develop a safe and reasonable treatment plan for patients. In addition, the system contains characteristic data, which can be obtained easily and quickly without expensive advanced technology and equipment. Allowing them to be used in communities or small hospitals with limited resources will also help areas with insufficient medical resources.

The predictive model used features from five clinical aspects: hemoglobin, history of atrial fibrillation, fast blood sugar, time to onset, and INR. Chang JY et al. proved in the paper that hemoglobin level is related to composite vascular events [39], and our experiment also selected it as a predictor, indicating that hemoglobin is closely related to stroke and bleeding conditions. A study from Xi’an Jiaotong University showed that atrial fibrillation may be an independent risk factor for hemorrhagic transformation in patients with acute ischemic stroke after intravenous thrombolysis with recombinant tissue plasminogen activator [40]. Tsinghua University Medical Center explored the effect of blood glucose indicators on outcomes after intravenous thrombolysis in acute ischemic stroke, suggesting that elevated blood glucose levels may lead to poor prognosis [41]. It can be seen in Figure 7 that the higher the blood glucose value, the higher the possibility of HC, which is consistent with this. In the related research of Yibing Chen [42], INR is a vital reference factor in stroke treatment. As for the time of onset, the results show that the shorter the time from onset to treatment, the more likely it is to cause HC, which may be inconsistent with the concept of treatment as soon as possible in actual treatment. The model contains 13 image features: features based on gray level co-occurrence matrix (GLCM) (6), features based on gray level correlation matrix (GLDM) (1), features based on gray run length matrix (GLRLM) (3), first-order statistics (3), see Figure 6; Figure 7. The original first-order 90 quantile is the most important feature in the model, which is similar to the result of Song Z [43], which shows that the experimental results have certain reliability.

Feng Wang et al. [44], also used machine learning to conduct a post-thrombotic HC study in multicenter data, with an AUC value of 0.82. Our research makes up for the lack of CT scanning image features and achieves better prediction results, which shows that combining various features can help improve the performance of the model. At present, more and more research on multimodality also supports this point. The ACC value of the XGB model on the internal validation cohort is 0.8554, and the AUC value is 0.9454. Compared with the previous related studies [16,17,18,19], the accuracy and generalization ability have been improved to a certain extent. At the same time, the ACC value of my model in the external validation cohort is 0.8431, and the AUC value is 0.9142, indicating that the generalization ability of the prediction model is good. In addition, when building the model, we did not ultimately reduce the feature dimension to a very low level, but when building the model, different models determined the appropriate number of features, to better play the model’s ability. Then, comprehensively compare the performance of the models through AUC, Delong test and other indicators, and obtain a model with better performance, which makes the results have a more reliable basis.

Based on artificial intelligence technology’s development, CDSS has been promoted and applied to a certain extent [45]. Concerning relevant systems, we have developed a prototype system using the final prediction model and python language to facilitate practical application and further verify our model’s capability. Our system adopts B/S architecture and places the calculation of the model on the server side, reducing the dependence on the hardware of the application environment. At the same time, it has high integration and good portability. The system can be used as a module in the hospital application system and can be conveniently used on the website to view the analysis results.

Our research also has several shortcomings, which are left for future studies. First, using image comics will cause losses when using software to extract image features. Next, we consider using deep learning to complete prediction tasks, which is expected to achieve better performance. Secondly, we used multiple central data in the experiment, but the research population and data were limited. The ML model is wholly based on past human behavior to predict. Therefore, the ability to continue learning to use the prediction model is also crucial. We plan to use new samples to adjust the model parameters for a while, hoping to improve the application’s performance continuously. Finally, if the CDSS prototype is not perfect and still needs further improvement, then actively promote the application, and analyze the system capability and acceptance capability.

## 5. Conclusions

In conclusion, we have established four prediction models based on multi-modal data from a multi-medical-center. After evaluating different indicators, the XGB model performs better than other models. The CDSS prototype is developed based on the prediction model using python tools. The system will provide doctors with a diagnostic basis, which is expected to reduce hemorrhagic complications and enable patients to obtain a good prognosis.

## Figures and Tables

**Figure 1 jpm-12-02052-f001:**
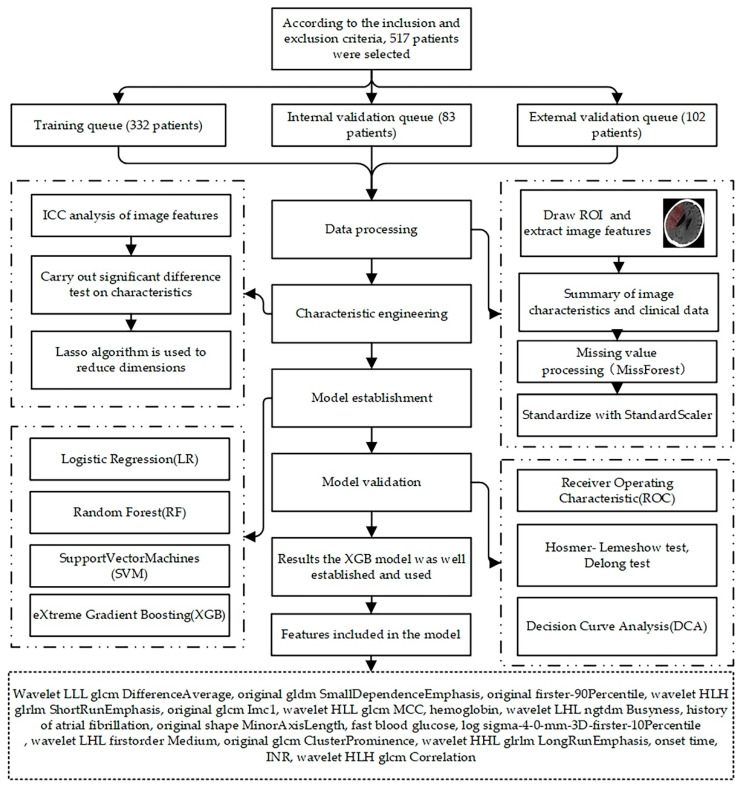
The overall flow of the model establishment.

**Figure 2 jpm-12-02052-f002:**
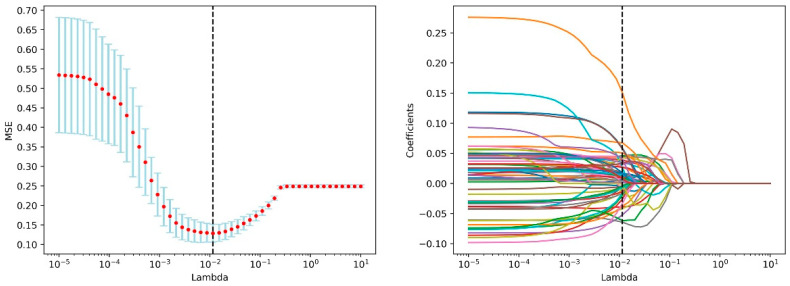
This is the result graph of Lasso algorithm, the graph on the (**left**) shows the curve of MSE loss changing with Lambda value, and the (**right**) graph shows the curve of characteristic weight coefficient changing with Lambda value.

**Figure 3 jpm-12-02052-f003:**
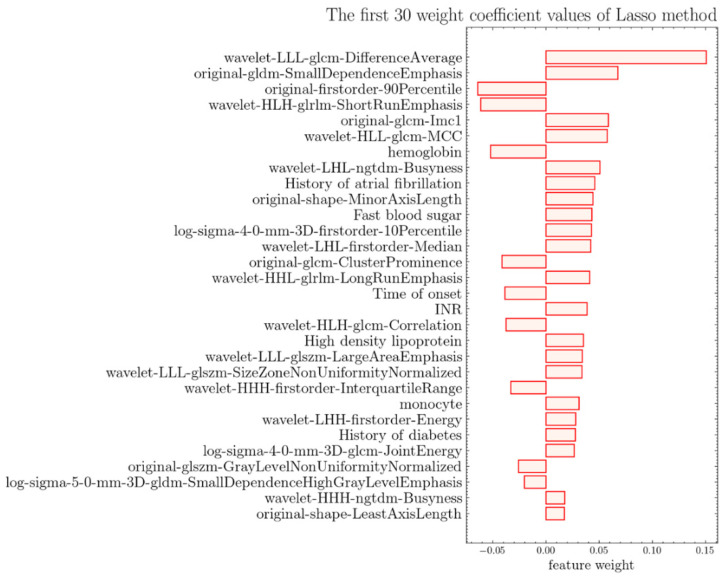
Bar chart of feature weight, showing the top 30 features of weight after Lasso algorithm screening.

**Figure 4 jpm-12-02052-f004:**
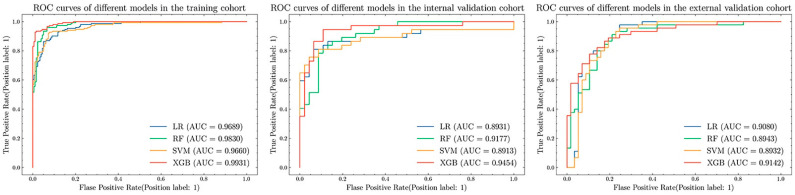
ROC curves of different models, from left to right, are the results of training set, internal verification set and external verification set.

**Figure 5 jpm-12-02052-f005:**
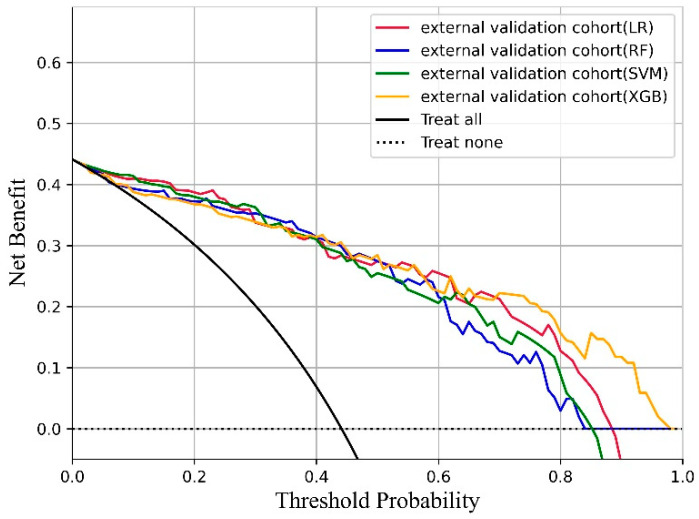
Decision curve of each model on external verification set.

**Figure 6 jpm-12-02052-f006:**
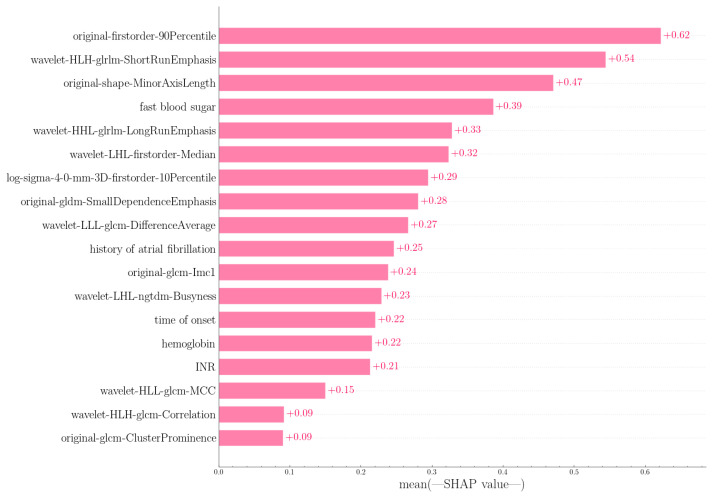
Average effect of features on model output amplitude.

**Figure 7 jpm-12-02052-f007:**
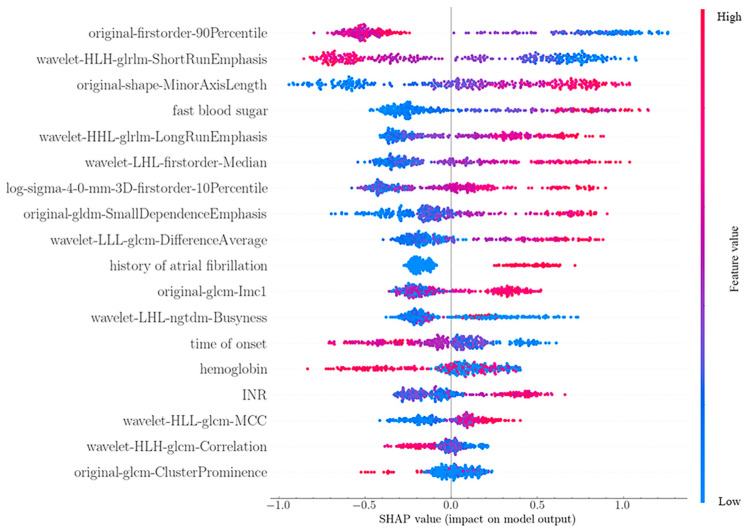
Snap summary diagram, the image degree of each feature to the model result, each row represents a feature, and the abscissa is the log level snap value. One point represents one sample, and the color represents the eigenvalue (red high, blue low).

**Figure 8 jpm-12-02052-f008:**
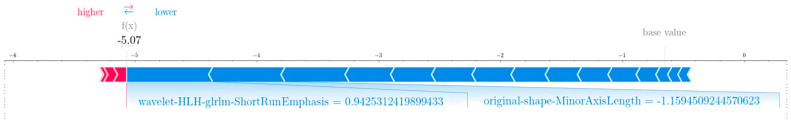
Example of shap value visualization diagram for negative samples, it shows that each feature has its own contribution, pushing the prediction result of the model from the base value to the final model output; Red indicates the features that will be predicted to be higher, and blue indicates the features that will be predicted to be lower.

**Figure 9 jpm-12-02052-f009:**
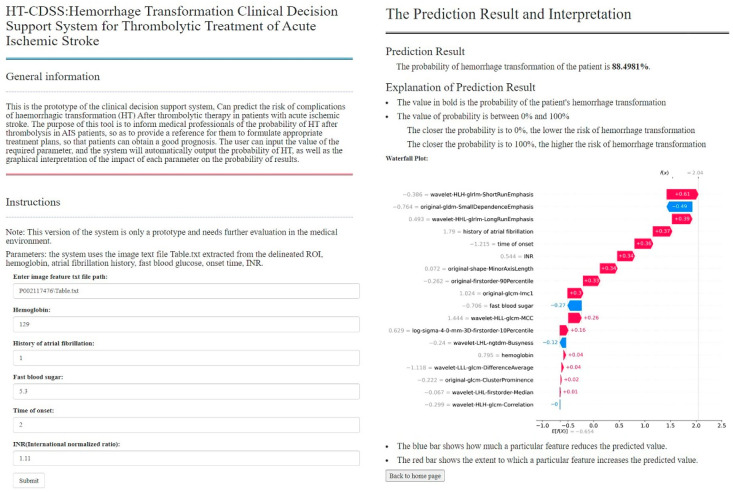
Clinical decision support system interface, the left is the input parameter page, and the right is the result display page.

**Table 1 jpm-12-02052-t001:** Baseline characteristics of training and internal validation cohorts.

Variables	Train (*n* = 332)	Internal (*n* = 83)	*p*
Demographic			
gender, *n* (%)			1
Male	206 (62.0)	51 (61.4)	
Female	126 (38.0)	32 (38.6)	
Age, years, mean ± SD	67.39 ± 12.44	66.18 ± 13.53	0.438
Past medical history			
Atrial fibrillation, *n* (%)			0.465
No-atrial-fibrillation	253 (76.2)	67 (80.7)	
Atrial-fibrillation	79 (23.8)	16 (19.3)	
Clinical manifestation			
Onset time, hours, mean ± SD	5.41 ± 2.81	5.43 ± 2.56	0.948
Laboratory examination			
Hemoglobin, g/L, mean ± SD	78.84 ± 63.21	75.49 ± 65.83	0.669
Monocytes, 10^9^/L, mean ± SD	0.55 ± 0.25	0.71 ± 1.21	0.028
Fast blood sugar, mean ± SD	7.83 ± 3.59	8.05 ± 3.14	0.606
INR, mean ± SD	1.03 ± 0.15	1.00 ± 0.12	0.152

Categorical variables are represented by the number (percent), and continuous variables are represented by mean (±standard deviation); INR, international normalized ratio.

**Table 2 jpm-12-02052-t002:** Performance of different models on different data sets.

Classifier	LR			RF			SVM			XGB		
	Training Cohort	Internal Validation Cohort	Validation Cohort	Training Cohort	Internal Validation Cohort	Validation Cohort	Training Cohort	Internal Validation Cohort	Validation Cohort	Training Cohort	Internal Validation Cohort	Validation Cohort
ACC	0.8976	0.8675	0.8333	0.9277	0.8554	0.8333	0.9187	0.8434	0.8137	0.9458	0.8554	0.8431
AUC	0.9689	0.8931	0.9080	0.9830	0.9177	0.8943	0.9660	0.8913	0.8932	0.9931	0.9454	0.9142
SEN	0.9150	0.7838	0.8000	0.9739	0.8108	0.8667	0.9346	0.7568	0.7778	0.9608	0.7568	0. 8222
SPE	0.8827	0.9348	0.8596	0.8883	0.8913	0.8070	0.9050	0.9130	0.8421	0.9330	0.9348	0.8596
PPV	0.8696	0.9062	0.8182	0.8817	0.8571	0.7800	0.8938	0.8750	0.7955	0.9245	0.9032	0.8222
NPV	0.9240	0.8431	0.8448	0.9755	0.8542	0.8846	0.9419	0.8235	0.8276	0.9653	0.8269	0.8596
F1	0.8917	0.8406	0.8090	0.9255	0.8333	0.8211	0.9137	0.8116	0.7865	0.9423	0.8235	0.8222

**Table 3 jpm-12-02052-t003:** Delong test results on training set between models.

Delong Test								
Classifier	LR		RF		SVM		XGB	
	z	*p*	z	*p*	z	*p*	z	*p*
LR	0.0000	1.0000	2.3144	0.0206	−0.6490	0.5164	3.7309	0.0002
RF	−2.3144	0.0506	0.0000	1.0000	−2.2130	0.0269	2.0787	0.0376
SVM	0.6490	0.5164	2.2130	0.0269	0.0000	1.0000	3.4741	0.0005
XGB	−3.7309	0.0002	−2.0787	0.0376	−3.4741	0.0005	0.0000	1.0000

Note: see Appendix A for the results of internal verification and external verification.

## Data Availability

The data used in this study are not publicly available due to restrictions in the data-sharing agreement.

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
