# Peer review of "Prediction of Hemorrhagic Complication after Thrombolytic Therapy Based on Multimodal Data from Multiple Centers: An Approach to Machine Learning and System Implementation"

_jpm, 2022, doi:10.3390/jpm12122052_

Round 1
Reviewer 1 Report
The article "Prediction of hemorrhagic transformation after thrombolytic 2 therapy based on multimodal data from multiple centers: an 3 approach to machine learning and system implementation" is extremely well written and thoroughly done. Other than minor text errors, I strongly recommend publication of this article in current form. Thank you for the review opportunity.
Reviewer 2 Report
General comments:
This article describes the development of a machine learning system, based on various statistical approaches, which is proposed to predict the hemorrhagic risk in patients treated with thrombolytics for acute stroke events. It concerns a study performed on a wide cohort of 517 patients (with 3 different inclusion criteria), analyzed for many biological parameters, all used for statistical analysis and filling data for the machine learning algorithm. The performances of each statistical method used are compared and authors conclude that the eXtreme Gradient Boosting provides the best results. This investigation is within the present tendencies to use statistics, algorithms and artificial intelligence to strengthen the knowledge and to develop tools for predicting the occurrence of disease complications, which then allows adjusting the patient management to reduce or to avoid critical clinical complications. Machine learning based on all patient data generated is then a key tool for that objective. This article is well documented and illustrated, but it contains too many complex equations and too many complicated figures are presented. They are not obvious or instructive enough for readers who are not highly specialized in that field. In addition, some definitions are well known and documented for epidemiological studies and statistics, and do not need to be explained in detail (as for example, the Positive Predictive Value or the Negative Predictive Value). Part of the methods' chapter can therefore be simplified.
Specific comments:
Some wordings need to be adjusted to the current medical practice.
In title, the wording "hemorrhagic transformation" does not look appropriate, and should be better reported as "hemorrhagic complication, or risk, or tendency or evolution". The same comment prevails throughout the text.
All abbreviations used must be fully written at their first citation; this is not the case for example in the abstract, or for "MRI" line 91.
On pages 4 and 5, is it necessary to show and to detail all the equations presented? Describing the 4 statistical models used, and referencing them, should be enough for the right understanding.
For the statistical analysis, I recommend using the current wordings: for example Negative Predictive Value (NPV) in place of Negative Recall, and the same for Positive Predictive Value (PPV).
Statistics' description can be simplified and shortened, as many terms are current ones. Equations page 6 are not necessary.
On table 1, some reported values require attention. For example, mean DBP looks too high comparatively to SBP (102.33 versus 116.80).
On this table 1, only the most significant biological variables need to be reported. Those with no statistic significant changes can just be mentioned in the text.
What are the meanings for the data reported for smoking or drinking habits? Figures are not easily understandable and SDs are huge.
The units used for blood cells must be indicated (for example G/L for platelets).
Can figures 2 and 3 be combined in a simple figure? Is the Lasso diagram of figure 2, right, useful in this article?
On page 9, figure 4 is missing.
Can figures 6, 7 and 8 be combined in a single figure?
Although globally acceptable, English language requires attention for some approximative wordings or syntax errors.
Reviewer 3 Report
The authors performed an interesting study on “Machin learning based prediction of hemorrhagic transformation after thrombolytic therapy using multimodal data”, however, a major revision is required before the decision:
1. The quality of some figures is very poor: Figures 6, 7, 8.
2. There is a description under Figure 1, as a part of this figure which is unclear. There is no explanation or reference to this figure in the text.
3. In Table 1, for demographic variables, “Male 206 ±62.0” what mean? It should be 206 (62.0%)
4. Line 126: “significantly better than methods such as KNN and MI”, what is MI? These are field-specific, non-standard abbreviations that the author must define in their first introduction to the manuscript.
5. There are inconsistencies between verb tenses in the manuscript like “line 61: Liu Z et al. reach AUC of 0.885 for prediction of HT”, “line 139: We obtain the best”.
6. Some parts of the manuscript should be rewritten because of grammatical errors or being unintelligible
- Line 156: We can explore similar the linear setup for classification
- Line 183: it can be applied to deal with data sparsity is a common problem in the real world
- Line 376: which our study makes up for.
- Line 376: One deficiency, and achieved good prediction results
7. The machine learning models were not explained well, LR model was described in detail with equations, while the concept of other models was not clearly explained. This part includes too long sentences (like lines 15l-155), non-informative content (like “XGBoost algorithm is more than 10 times faster than popular models used in machine learning (ML) and deep learning (DL)… which popular models?). This part should be totally rewritten.
8. In the results section, it is better to bold the best performance at each table to let the reader compare the methods easily so that the reader does not have to search for related information.
9. Please cite this paper in your manuscript. https://www.nature.com/articles/s41598-021-89352-8
Reviewer 4 Report
This is an innovative study leading to the development of prediction tools for hemorrhagic transformation in patients undergoing intravenous thrombolytic therapy.
Strengths:
- This is a first of its kind study to build a prediction system for hemorrhagic transformation after thrombolytic treatment of acute ischemic stroke in patients, thus enabling patients to receive proper prognosis as well as reduce bleeding complications.
- The references are most recent and comprehensive.
- Interpretation and presentation of previous studies is accurate.
- No major suggestions for improvements.
- Clarity and context in this paper are good.
Minor comments:
- -Abbreviations like CDSS are used before providing the expanded form or what they stand for.
- Language editing and proper punctuation is needed.
Round 2
Reviewer 3 Report
The authors provided my corrections and I recommend the manuscript for publication.